# Short Peptides of Innate Immunity Protein Tag7 Inhibit the Production of Cytokines in CFA-Induced Arthritis

**DOI:** 10.3390/ijms232012435

**Published:** 2022-10-17

**Authors:** Georgii B. Telegin, Aleksandr S. Chernov, Alexey N. Minakov, Irina P. Balmasova, Elena A. Romanova, Tatiana N. Sharapova, Lidia P. Sashchenko, Denis V. Yashin

**Affiliations:** 1Institute of Bioorganic Chemistry (RAS), 117997 Moscow, Russia; 2Moscow State University of Medicine and Dentistry, 127473 Moscow, Russia; 3Institute of Gene Biology (RAS), 119334 Moscow, Russia

**Keywords:** mice, CFA, inflammation, arthritis, TNFα, Tag7, 17.1 and 17.1a peptides

## Abstract

The pathogenesis of autoimmune arthritis is a hot topic in current research. The main focus of this work was to study cytokines released in CFA-induced arthritis in ICR mice as well as the regulation of blood levels of cytokines by two peptides of the innate immunity protein Tag7 (PGLYRP1) capable of blocking the activation of the TNFR1 receptor. Arthritis was induced by local periarticular single-dose injections of 40 µL of complete Freund’s adjuvant (CFA) into the left ankle joints of mice. The levels of chemokines and cytokines in plasma were measured using a Bio-Plex Pro Mouse Cytokine Kit at 3, 10, and 21 days after arthritis induction. Tag7 peptides were shown to decrease the blood levels of the pro-inflammatory cytokines IL-6, TNF, and IL-1β. Administration of peptides also decreased the levels of chemokines MGSA/CXCL1, MIP-2α/CXCL2, ENA78/CXCL5, MIG/CXCL9, IP-10/CXCL10, MCP-1/CCL2, and RANTES/CCL5. Furthermore, a decrease in the levels of cytokines IL7, G-CSF, and M-CSF was demonstrated. Addition of the studied peptides strongly affected IFN-γ concentration. We believe that a decrease in the levels of cytokine IFN-γ was associated with a therapeutic effect of Tag7 peptides manifested in alleviation of the destruction of cartilage and bone tissues in the CFA-induced arthritis.

## 1. Introduction

The study of the pathogenesis of autoimmune arthritides, including rheumatoid arthritis, is mostly based on experimental animal models. Cytokines play a key role in the pathogenesis of rheumatoid arthritis. They are present in the complex process of disease initiation and orchestrate the common phenotype of persistent synovial hyperplasia, immune cell infiltration, and joint destruction that ensues [1]. Most commonly, the pool of cytokines at the first stage of disease is formed in the setting of inflammatory processes [2].

Pro-inflammatory cytokines are involved in the development of these inflammatory processes. Cytokines IL-1β, IL-6, and TNF often play a key role [3]. IL-6 is one of the principal mediators of the acute phase of inflammation. It stimulates the proliferation and differentiation of B and T cells as well as leukopoesis [4]. Cytokine IL-1β is produced by cells in an inactive form as part of the intracellular molecular complex, inflammasome, and is activated after release from the cell during apoptosis [5]. At subsequent stages, cytokines of several different groups are involved in the development of pathological processes. An important role in the development of chronic inflammation is played by the colony-stimulating factors (CSF), such as macrophage CSF (M-CSF) and granulocyte CSF (G-CSF), which maintain proliferation and activation of the main cells of an immune system producing pro-inflammatory cytokines. It has been shown that, in mice with arthritis, G-CSF levels are increased both in the serum and inflamed extremities, and that a G-CSF blockage results in a significant decrease in the severity of arthritis and reduction in blood neutrophil counts [6,7].

To date, it has been shown that osteoclasts play an important role in the development of articular bone erosion in adjuvant-induced arthritis, and M-CSF is recognized as a key factor responsible for survival and proliferation of the osteoclast progenitor cells [8,9]. An important role in the proliferation of immune cells is played by IFN-γ, which triggers the activation of natural killer (NK) cells and lymphocytes [10]. The depletion of the non-activated lymphocyte pool results in a production of growth factors and chemokines stimulating the migration of new cells to a site of autoimmune response and their consequent proliferation [11]. The resulting pathological immune response is dependent on the balance between activating cytokines and anti-inflammatory cytokines. Among the latter, IL-10 and IL-4 play an important role [12]. IL-4 takes part in the launch of humoral immune responses, contributing to the differentiation of naive helper T cells into Th2. Upon the IL-4-mediated activation, Th2 subsequently produce additional IL-4. Experiments have shown that, in collagen-induced arthritis, IL-4 suppresses the production of IL-17 [13], thus reducing the intensity of inflammatory response.

A particular role in the progress of autoimmune arthritis is played by TNF [14]. This cytokine has been well studied, and it was demonstrated that, in addition to its ability to induce the production of other cytokines, it possesses other activities, including an ability to directly induce programmed cell death [15]. TNF exerts its activity by binding to a specific TNFR1 receptor. We have shown previously that the innate immunity protein Tag7 is able to bind directly to a TNF receptor, the TNFR1 protein, and to inhibit signaling through this receptor [16]. The mammalian gene encoding Tag7 was initially discovered at our institute. Subsequently, it was also discovered in insects and named PGRP-S (PGLYRP1). [17,18]. In insects, the protein encoded by this gene plays an important role in the immune response because it directly binds to peptidoglycans on the bacterial wall and, via interaction with the Spätzle protein, transmits a signal directly to the Toll receptor [19]. In mammals, there are several Toll receptors that interact directly with various types of pathogens, including peptidoglycans; however, the Tag7 protein retains its important function of activating the immune system [20].

We have shown that the Tag7 protein is involved in the activation of TREM-1 receptor that serves as an enhancer of the pro-inflammatory In review Running Title 3 signal induced by Toll receptors [21]. Accordingly, in mammals, Tag7 also serves as a co-activator of the cascade of immune reactions induced through the Toll receptor. We have identified a site in Tag7 that is responsible for binding to TNFR1 and have isolated a receptor-binding peptide [22,23]. This peptide, 17.1, and its truncated variant, 17.1a, have the potential to inhibit cytotoxic signal transmission through the TNFR1 receptor [23]. Recently, it has been shown that administration of peptide 17.1 to mice with the CFA-induced experimental arthritis has a protective effect against cartilage and bone destruction [24,25].

This work was aimed at identification of cytokines that are involved in pathological processes in the CFA-induced arthritis and what are the effects of the 17.1 peptide (TNFR1 inhibitor) and its truncated variant, 17.1a, on the levels of these cytokines.

## 2. Results

### 2.1. Pro-Inflammatory Cytokines

We used a mouse model of adjuvant-induced arthritis, and at early time-points of the experiment, levels of “classical” pro-inflammatory cytokines, such as IL-1β, IL-6, and TNF, were measured. The levels of these cytokines in the serum of experimental animals against the controls (intact animals) are shown in Figure 1 (see Appendix A) at the time-points from the start of the study.

From the provided results, it can be seen that CFA administration results in an increase in the levels of three pro-inflammatory cytokines (IL-1β, IL-6, and TNF). An increase in IL-6 levels was the most pronounced. Without any treatment, the levels of this cytokine reached their peak values (3.5 times above baseline) by study day 10, after which they decreased but still remained about two times higher than the control values. With Norocarp, this increase declined during the entire follow-up period. The study peptides 17.1 and 17.1a (the truncated variant) exerted inhibitory activity on day 21, and on days 10 and 21, respectively. TNF was the second most elevated cytokine. With CFA treatment, serum levels of this cytokine also reached their peak values (two times above the baseline) by day 10. An increase in TNF levels by day 10 was inhibited by the addition of both Norocarp and the studied peptide 17.1a. On day 21, the TNF level returns to subnormal. IL-1β is produced by cells in an inactive form and is activated when released during apoptosis [5]. From this point of view, an analysis of the effects of these peptides on IL-1β levels warrants investigation of the mechanisms of their action and their relation to apoptosis.

The IL-1β levels demonstrate a rather high increase (by 5.4-fold) on day 21, which suggests a significant activation of apoptosis during this period. All tested agents demonstrated a statistically significant reduction in IL-1β levels, with peptide 17.1a and Norocarp achieving reduction on day 21, and peptide 17.1 on days 10 and 21. The levels of IL-1α and IL-17A did not change in the setting of CFA-induced arthritis.

### 2.2. Chemokines

When analyzing the results of measuring chemokines in the serum of animals exposed to different agents, all chemokines were divided into two groups according to their structure and mechanism of action on the cells of immune system. The first group consisted of CXCL chemokines with an effect on neutrophilic granulocytes and lymphocytes, while the second group consisted of CCL chemokines having effects on monocytes, lymphocytes, eosinophils, and basophils. The first group included the following chemokines: MGSA/CXCL1 (peptide with melanoma growth stimulatory activity), MIP-2α/CXCL2 (macrophage inflammatory protein 2α), ENA78/CXCL5 (epithelial cell-derived neutrophil-activating protein-78), MIG/CXCL9 (monokine induced by gamma interferon), and IP-10/CXCL10 (interferon gamma inducible protein 10) (Figure 2) (see Appendix A).

Among the CXCL chemokines, IP-10/CXCL10 demonstrated the highest degree of deviation from the controls in all groups of animals, with levels increased above the controls. Without exposure to any agents in the setting of mouse experimental arthritis, the levels of this cytokine were increased 2.2-fold already on day 3, and stayed close to this level over the next 3 weeks. In the mice treated with Norocarp, the levels of this cytokine were decreased by day 10, whereas in mice treated with both study peptides, its levels were decreased by day 21. MGSA/CXCL1 is the second chemokine of this group, the levels of which were the most increased after treatment with these agents. Without exposure to any agents, the levels of this chemokine were increased 1.8-fold already on day 3, after which they steadily decreased, reaching the control values by day 21. Norocarp caused a decrease in the levels of this chemokine on day 10. Peptide 17.1a reduced the levels of CXCL1 on days 3 and 10, while peptide 17.1 reduced them only on day 10. In the setting of this experiment, the levels of MIG/CXCL9 were moderately increased (1.3- to 1.7- fold) over the entire period of the experiment. Norocarp reduced the levels of this chemokine on day 10, peptide 17.1a on days 10 and 21, and peptide 17.1 only on day 21. Under the effect of CFA, the levels of MIP-2α/CXCL2 were increased on day 21; however, this increase was suppressed by all tested agents. The levels of ENA-78/CXCL5 were also increased on day 21, and this increase was suppressed by all tested agents. The second group of chemokines included MCP-1/CCL2 (monocyte chemoattractant protein-1) and RANTES/CCL5 (regulated upon activation, normal T cell expressed and secreted). The time course of serum levels of these chemokines in the setting of adjuvant-induced arthritis in different groups is shown in Figure 3 (see Appendix A).

Chemokines of this group play a particularly important role in adjuvant-induced arthritis in animals [26,27]. The levels of MCP-1/CCL2 on day 3 were the highest in untreated animals (5.8-fold above control) and in animals treated with peptide 17.1 (5.4-fold); the least pronounced increase (3.1-fold) was observed in animals treated with peptide 17.1a. Starting from day 10, the levels of this cytokine sharply decreased in all groups, except in mice treated with Norocarp, in which, on the contrary, an 8.7-fold increase in the levels of MCP-1/CCL2 was observed. The inhibitory effect was apparent only for peptide 17.1a, while peptide 17.1 and Norocarp exerted no therapeutic effect. The levels of RANTES/CCL5 demonstrated a rather high increase on day 10, while in mice treated with peptides 17.1 and 17.1a, they remained in a range close to the control values. Treatment with Norocarp reduced the levels of this cytokine only on day 21. The levels of chemokines MCP-1/227 CCL2, MIP-1α/CCL3, MIP-1β/CCL4, and eotaxin/CCL11 did not change by treatment with the tested peptides.

### 2.3. Hematopoietic Cytokines and Growth Factors

Another group of cytokines involved in the regulation of hematopoiesis consists of cytokines that function as colony-stimulating and growth factors (Figure 4) (see Appendix A). They include IL-7, granulocyte colony-stimulating factor (G-CSF), and macrophage colony-stimulating factor (M234-CSF). The levels of these factors measured during the experiment are summarized in Figure 4. In the group of mice with adjuvant-induced arthritis without exposure to any agents, lymphopoetin IL-7 demonstrated the highest degree of deviation from controls, with a 2.5-fold increase by day 10 and subsequent decrease to control values. Treatment with all agents resulted in a decrease in IL-7 levels, which was statistically significant for peptide 17.1a and, in particular, peptide 17.1.

Our studies on the mouse model of adjuvant-induced arthritis have shown that G-CSF levels were above the baseline during the entire period of study. Norocarp and 17.1a peptide resulted in a decrease in cytokine levels by day 10 and by day 21, respectively. M-CSF levels were increased by day 21; however, this increase was suppressed by all agents tested. The levels of IL-3, IL-5, IL-9, and LIF, as well as those of GM-CSF and VEGF, did not increase in the setting of CFA-induced arthritis.

### 2.4. Cytokines of the Effector Phase of Immune Response

The measured levels of cytokines that induce cellular (cytotoxic) immune reactions, such as IFN-γ inducing cellular (cytotoxic) immune response and IL-4 involved in the induction of humoral immune response, in the setting of experiments are shown in Figure 5 (see Appendix A). The levels of IFN-γ, one of the core components of the effector phase of cellular response in autoimmune diseases, progressively increased during the entire study period. The most pronounced suppression of this increase by all agents was observed on day 10.

IL-4 demonstrated an increase on days 10 and 21 of the study, but the level of this cytokine stays rather low. Peptide 17.1a and Norocarp demonstrated an inhibitory effect in both study periods. The levels of IL-2, IL-12p40, IL-12p70, and IL-15, as well as those of IL-10 and IL-13, did not increase in the setting of CFA-induced arthritis.

## 3. Discussion

The results obtained in this study permit drawing the following conclusions:(1)Under the effect of CFA, there is an increase in the levels of all pro-inflammatory cytokines, such as IL-1β, IL-6, and TNF, all chemokines tested (except CCL11), and IFN-γ, as well as IL-7, G-CSF, and M-CSF.(2)Peptide 17.1 and its truncated variant 17.1a caused a decrease in the levels of above-mentioned cytokines and chemokines, which was most pronounced by days 10 and 21. In our previous work, we have shown that administration of peptide 17.1 may reduce the effect of CFA-induced arthritis on mouse cartilage and bone tissue [24].

In the experiments conducted in that work, a protective effect of peptide 17.1 was most pronounced on day 21. In this work, we used blood samples from the mice from our previous work, so that we had the possibility to connect the clinical picture of our previous data, namely a strong reduction in inflammatory symptoms and a very strong reduction in damage to cartilage and bone tissue in the CFA-induced arthritis, to the concentrations of broad spectra of cytokines and chemokines. In this work, we have also demonstrated that peptides 17.1 and 17.1a exert the most pronounced effects on the levels of expressed cytokines and chemokines by days 21 and 10 of the study, respectively.

Earlier, we have also shown that peptide 17.1 could inhibit the production of cytokines IL-1β, IL-6, TNF, and IFN-γ expressed by human blood cells ex vivo upon induction with lipopolysaccharide. The results of the study described here confirm that the levels of pro-inflammatory cytokines IL-1β, IL-6, and TNF, as well as those of IFN-γ, an immune response initiator, that were increased upon induction by CFA are reduced after administration of peptides 17.1 and 17.1a. The list of CFA-induced cytokines, the levels of which are suppressed by administration of peptide 17.1 and its truncated variant 17.1a, has been expanded during this study. The expanded list includes chemokines CXCL1, CXCL2, CXCL5, CXCL9, CXCL10, as well as CCL2 and CCL5. IP-10/CXCL10 is known to be secreted upon induction with IFN-γ by neutrophils, eosinophils, monocytes, epithelial and endothelial cells, and stromal cells, as well as keratinocytes; this chemokine acts as a ligand of the CXCR3 receptor expressed by activated T and B cells, NK cells, dendritic cells, and macrophages. IP-10/CXCL10 triggers chemotaxis, apoptosis, and inhibition of cell growth and angiostasis [28]; thus, they are decreasing the intensity of the cellular immune response, which should be considered as a quite beneficial effect in autoimmune arthritides that usually undergo a cellular type of development [29]. MGSA/CXCL1 is produced by macrophages, neutrophils, epithelial cells, or Th17 cells, and its expression could be induced by IL-1, TNF, or IL-17. MGSA/CXCL1 acts as a chemoattractant for neutrophils or other non-hematopoietic cells, recruiting them to the site of damage or infection, and plays an important role in the regulation of immune and inflammatory reactions [30]. MIG/CXCL9 activity is manifested in response to IFN-γ and realized via the attraction of cytotoxic T-lymphocytes (CTL), NK cells, NK T cells, and macrophages [31].

Chemokine MIP-2α/CXCL2 is produced by mast cells and macrophages, induces the attraction of neutrophils, and often acts synergistically with MGSA/CXCL1 [32]. MIP-1α/CCL3 and RANTES/CCL5 demonstrated high constitutive expression on macrophages in adjuvant-induced arthritis, which correlated with the expression of CCR2 (MCP-1/CCL2 receptor) by macrophages and was proved to play an important role in maintaining inflammatory changes in the joints. At the same time, RANTES/CCL5 may act through another receptor, CCR3, the expression of which is suppressed in this experimental model [31]. In the clinical setting, a high degree of concordance has been demonstrated between the status of oxidative stress in rheumatoid arthritis and the levels of MIP-1β/CCL4 and MCP-1/CCL-2 [33].

In the study conducted here, we have also demonstrated a protective effect of the tested peptides against an increase in the levels of cytokines IL-7, G-CSF, and M-CSF. Observations made in our work suggest that peptides 17.1 and 17.1a prevent the transmittance of the TNF signal via the TNFR1 receptor [22,23], and exert inhibitory effects, reducing an increase in the levels of pro-inflammatory cytokines and those of the cytokines that initiate autoimmune responses. This activity may be one of the mechanisms accounting for their protective effect against the destruction of bone and cartilage tissue in the CFA-induced arthritis. The second mechanism may consist of direct inhibition of TNF-induced cell death, which is indirectly evidenced by a significant increase in the levels of IL-1β on day 21 and suppression of this increase by both peptides tested. We have noticed that both peptides significantly reduced the levels of IFN-γ, which are dramatically 26-fold increased during induction of arthritis. It is known that the choice of arthritis model is of paramount importance for studying the mechanisms of immune reactions as emphasized in current literature reviews devoted to this topic [9]. Currently, the most recognized are the models of arthritis induced by adjuvant [34], serum transfer [35], and tumor necrosis factor [36]. We have used a model where intra-articular changes were caused by a non-specific agent, complete Freund’s adjuvant. In this setting, reactions with preferential involvement of innate immune cells are likely to prevail. As emphasized in one recent work devoted to this topic, in this model, IFN-γ may play a key role in the extension of arthritis at the early stage. This cytokine produced by T-helper cells, CD4+, was involved in the induction of pro-inflammatory cytokines (TNF and Il-1β) and reprogramming the vascular network, which resulted in changes in vascular permeability, thus permitting autoantibodies to enter the target tissues [37].

With regard to our data, the pathogenetic role of IFN-γ was most apparent on day 10 of the experiment, with the degree of change being higher than for any other cytokine. In a group of mice not exposed to any additional agent, the levels of IFN-γ demonstrated a 3-fold increase on day 3 already, and, by day 10, they reached a peak value that was 8.8 times above the control values. By day 21 of the experiment, the levels of this cytokine decreased but still remained elevated (3.9-fold above control values). In other words, it is reasonable to believe that IFN-γ is one of the key players in the adjuvant-induced arthritis. Moreover, as mentioned earlier, all three agents significantly suppressed this predominant effect (in particular, peptide 17.1a).

The prevalence of rheumatoid arthritis is 0.5–1%, with a female:male ratio of 3:1 [38,39]. In the literature, this is explained by gender differences in the biological and molecular mechanisms underlying the pathogenesis of pain in arthritis [40,41]. At the same time, morphological differences in the development of CFA-induced arthritis in male and female mice are rather weakly expressed [42]. In our study, we present the results of using peptides 17.1 and 17.1a in the CFA-induced arthritis model in male ICR mice only. The use of female mice may produce different results because there are gender differences in rheumatoid arthritis [41,43].

## 4. Materials and Methods

### 4.1. Animals

Male ICR mice with an average weight (±SEM) 39.3 ± 1.65 g were used. All animals were housed under standard conditions for laboratory animals as specified by BIBC, RAS (the Unique Research Unit Bio-Model of the BIBCh, RAS; the Bioresource Collection—Collection of SPF Laboratory Rodents for Fundamental, Biomedical and Pharmacological, contract No. 075-15-2021-1067), which has an international accreditation from AAALACi. All experiments and manipulations were approved by the institutional animal care and use committee (IACUC protocol No. 713/20 from 06/08/20).

### 4.2. Experimental Groups

The animals were randomly assigned to five treatment groups with five animals in each group (*n* = 7): Group 1, control, intact animals; Group 2, induction of arthritis in animals that received intravenous injection of 100 μL of physiological saline; Group 3, induction of arthritis in animals that received peptide 17.1a (intravenous injection of 120 μg in 100 μL of physiological saline); Group 4, induction of arthritis in animals that received Norocarp (intravenous injection of 150 μg carprofen); and Group 5, induction of arthritis in animals that received peptide 17.1 (intravenous injection of 120 μg in 100 μL of physiological saline). The animals were followed up for 3, 10, and 21 days.

### 4.3. CFA-Induced Arthritis Model

Induction of arthritis was performed by a single injection of 40 μL complete Freund’s adjuvant (CFA) into the left ankle joint according to a method described earlier [22]. The study peptides and reference agents were intravenously injected 24 h after the induction of inflammation.

### 4.4. The Assay of Chemokines and Cytokines

The levels of chemokines and cytokines in mouse plasma were measured using Bio-Plex Pro Mouse Cytokine Panel 33-Plex kit (Bio-Rad, Hercules, California, USA). Blood was collected individually from the retro-orbital sinus at 3, 10, and 21 days after arthritis induction, into EDTA tubes to obtain plasma. Plasma samples diluted at 1:3 (50 μL) were incubated with magnetic capture beads, washed, and then incubated with the detecting antibodies and SA-PE. Data were obtained using a Luminex 200 analyzer and analyzed using xPONENT software, v.4.3. (Saluggia, Italy).

### 4.5. Peptides

The peptide synthesis was performed on an automated peptide synthesizer, Gilson Quad-Z. Purification of peptides was performed on a preparative chromatography system, HPLC Gilson, consisting of a gradient pump (322), detector (155), and a fraction collector (GX 271) under the control of Trilution^®^ software (v.4). Analysis of purity was performed on a chromatograph (Thermo Accela UPLC) with an ion trap mass spectrometer detector (Thermo Finnigan LCQ Deca XP Plus). The following chemicals were used: a block-copolymer carrier, Tentagel HL, with the terminal amino group modified with carboxy-trityl linker (Tentagel-TRT), Fmoc-protected amino acids (Iris Biotech), 4-methylpiperidine and collidine (Acros Organics), HATU (Sigma-Aldrich), and trifluoracetic acid (Solvay).

### 4.6. Statistical Analysis

Data were analyzed using Statistica software (StatSoft^®^, v.12.6). The results are presented as mean ±SD. Statistically significant differences were determined using *t*-tests. A value of *p* < 0.05 was considered to be statistically significant, compared with the CFA+saline group. The plate-specific intra-assay coefficient of variation (CV) for each marker was calculated as follows: (standard deviation (SD) of the duplicate values/mean of the duplicate values) ×100 (see Appendix A).

## 5. Conclusions

Thus, it should be acknowledged that all three agents (peptides 17.1a and 17.1, as well as Norocarp) exhibit some immunomodulatory activity, which is most potent in the peptide 17.1a. This activity is mainly manifested in the prevention of a high increase in the levels of a key cytokine, IFN-γ. Furthermore, as was shown in the final stages of the study, this agent does not produce some adverse effects typical for this model: it does not result in a diagnostically meaningful decrease in lymphopoetins and IL-10 with its immunosuppressive and anti-inflammatory effects and does not cause an increase in the levels of IL-17A and TNF, which are the main signs of a chronic process.

## Figures and Tables

**Figure 1 ijms-23-12435-f001:**
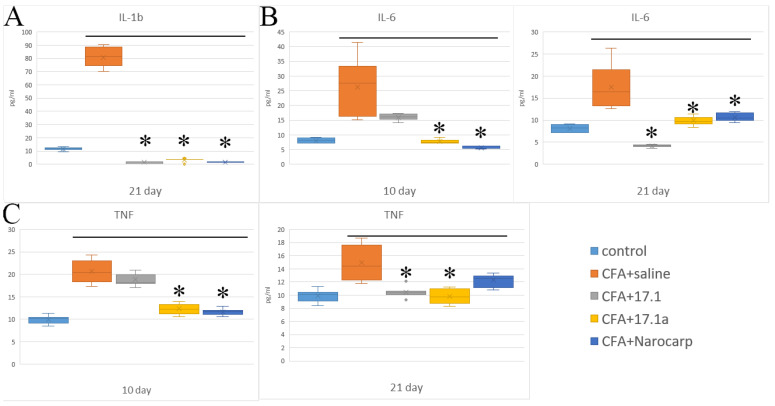
Changes in the level of pro-inflammatory cytokines ((**A**) IL-1b; (**B**) IL-6; (**C**) TNF) in the plasma 145 of mice after CFA-induced arthritis. (Control—control animals without CFA injection, CFA—animals with induced via CFA arthritis, + saline—animals were treated by phosphate buffered saline [PBS], +17.1a, +17.1, +Norocarp—animals were treated by a specified agent; *n* = 7 for each group). (*p*-value: * < 0.05).

**Figure 2 ijms-23-12435-f002:**
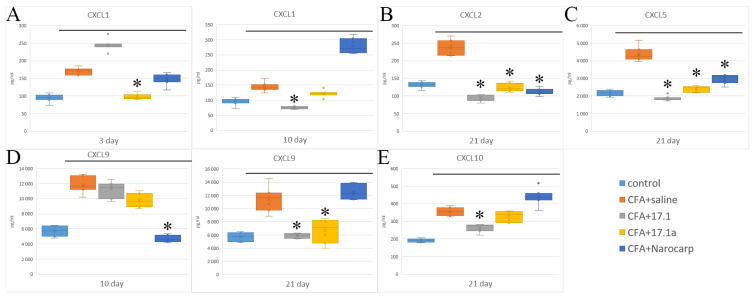
Changes in the level of chemokines ((**A**) CXCL1; (**B**) CXCL2; (**C**) CXCL5; (**D**) CXCL9; (**E**) CXCL10) in the plasma of mice after CFA-induced arthritis. (Control—control animals without CFA injection, CFA—animals with induced via CFA arthritis, +saline—animals were treated by phosphate buffered saline (PBS), +17.1a, +17.1, +Norocarp—animals were treated by a specified agent, *n* = 7 for each group). (*p*-value: * < 0.05).

**Figure 3 ijms-23-12435-f003:**
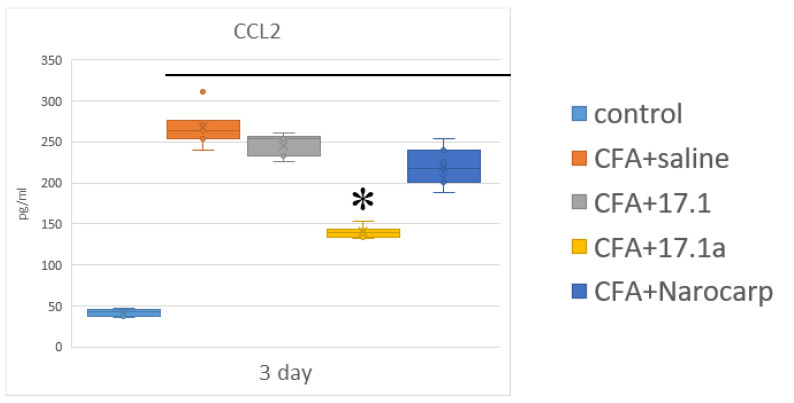
Changes in the level of CCL2 chemokine (CCL2) in the plasma of mice after CFA212 induced arthritis. (Control—control animals without CFA injection, CFA—animals with induced via CFA arthritis, +saline—animals were treated by phosphate buffered saline (PBS), +17.1a, +17.1, +Norocarp—animals were treated by a specified agent, *n* = 7 for each group). (*p*-value: * < 0.05).

**Figure 4 ijms-23-12435-f004:**
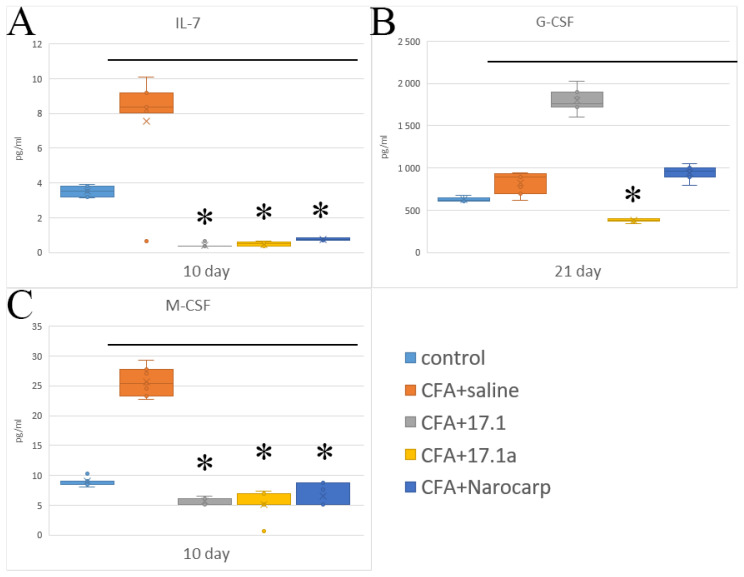
Changes in the level of hematopoietic cytokines ((**A**) IL-7) and growth factors ((**B**) G-CSF; ((**C**) M-CSF) in the plasma of mice after CFA-induced arthritis. (Control—control animals without CFA injection, CFA—animals with induced via CFA arthritis, +saline—animals were treated by phosphate buffered saline (PBS), +17.1a, +17.1, +Norocarp—animals were treated by a specified agent, *n* = 7 for each group). (*p*-value: * < 0.05).

**Figure 5 ijms-23-12435-f005:**
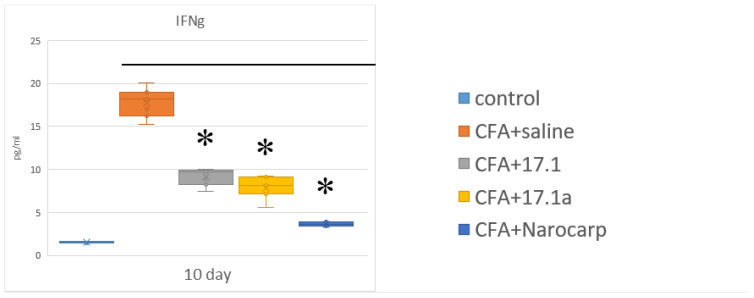
Changes of the level of cytokines in the effector phase of immune response (IFN-γ) in the plasma of mice after CFA-induced arthritis. (Control—control animals without CFA injection, CFA—animals with induced via CFA arthritis, +saline—animals were treated by phosphate buffered saline (PBS), +17.1a, +17.1, +Norocarp—animals were treated by a specified agent, *n* = 7 for each group). (*p*-value: * < 0.05).

## Data Availability

Not applicable.

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
