# Peer review of "Short Peptides of Innate Immunity Protein Tag7 Inhibit the Production of Cytokines in CFA-Induced Arthritis"

_ijms, 2022, doi:10.3390/ijms232012435_

Round 1

Reviewer 1 Report

Dear Autor,

Well designed research with clearly written English. 

Kind regards,

DRagana 

Author Response

The authors would like to thank the reviewer for all useful and helpful comments on our manuscript.

Reviewer 2 Report

This paper by Telegin et al. describes the effect of Tag7 peptides in the production of cytokines and chemokines in the CFA model of arthritis.

Previous publications by the authors described the reduction of clinical symptoms and histological characteristics of the disease using those peptides.

This present extends the previous ones and measures cytokines and chemokines present in the plasma of mice.

The data give significance to the previous studies but are not of utmost originality.

Are error bars SEM or SD? Data are representing values from different animals, yet the variation seems very low. How is this explained?
How many mice are in each group? Figure 1 indicates 145 plasma but the number of mice per group is not indicate.

Data should be best presented in the form of grouped graphs showing individual dots (mice) + bars. 

In some instances, data are below biological significance (IL-4, IL-7).

Author Response

The authors would like to thank the reviewer for all useful and helpful comments on our manuscript. All comments have been taken into account and the paper has been revised accordingly. Please see article revisions the attachment.

Answers to Comments and Suggestions for Authors:

1) The data give significance to the previous studies but are not of utmost originality.

In our previous studies we have demonstrated the therapeutic effect of peptides of innate immunity protein Tag7. Our data in this manuscript show that these peptides not only inhibit TNF-TNFR1 interaction but also change the spectrum of cytokines produced in mice. This data allow us to suggest that blocking TNF-TNFR1 interaction via studied peptides changes concentration of many cytokines and chemokines and strongly affects autoimmune response during induced arthritis. This investigation not only broadens the list of affected molecular pathways, but also gives new insights on the participation of various signaling molecules in system inflammation in mouse model.

2) Are error bars SEM or SD?

error bars is SD. see supplements tables

3) Data are representing values from different animals, yet the variation seems very low. How is this explained? How many mice are in each group?

n = 7 for each groups. We have reworked all figures in response to your comments and presented the data in form of Box-Whisker plots. In some cases the variation was rather high, but in most of this cases we do not uncover the reliable difference in results.

4) Figure 1 indicates 145 plasma but the number of mice per group is not indicate.

typo, corrected

5) Data should be best presented in the form of grouped graphs showing individual dots (mice) + bars.  In some instances, data are below biological significance (IL-4, IL-7).

We have reworked all figures in response to your comments and presented the data in form of Box-Whisker plots. We agree with the reviewer that concentration of IL-4 was rather low to make biologically significant conclusion and we have removed these data from the figure 5 and the text of the manuscript.
